# Prehospital stratification in acute chest pain patient into high risk and low risk by emergency medical service: a prospective cohort study

Kristoffer Wibring ![ORCID],[1,2] Markus Lingman,[3,4] Johan Herlitz,[5] Sinan Amin,[6] Angela Bång[1]

► Prepublication history and supplemental material for this paper is available online. To view these files, please visit the journal online (http://dx.doi.org/10.1136/bmjopen-2020-044938).

For numbered affiliations see end of article.

**Correspondence to**
Kristoffer Wibring;
kristoffer.wibring@gu.se

## ABSTRACT

**Objectives** To describe contemporary characteristics and diagnoses in prehospital patients with chest pain and to identify factors suitable for the early recognition of high-risk and low-risk conditions.

**Design** Prospective observational cohort study.

**Setting** Two centre study in a Swedish county emergency medical services (EMS) organisation.

**Participants** Unselected inclusion of 2917 patients with chest pain contacting the EMS due to chest pain during 2018.

**Primary outcome measures** Low-risk or high-risk condition, that is, occurrence of time-sensitive diagnosis on hospital discharge.

**Results** Of included EMS missions, 68% concerned patients with a low-risk condition without medical need of acute hospital treatment in hindsight. Sixteen per cent concerned patients with a high-risk condition in need of rapid transport to hospital care. Numerous variables with significant association with low-risk or high-risk conditions were found. In total high-risk and low-risk prediction models shared six predictive variables of which ST-depression on ECG and age were most important. Previously known risk factors such as history of acute coronary syndrome, diabetes and hypertension had no predictive value in the multivariate analyses. Some aspects of the symptoms such as pain intensity, pain in the right arm and paleness did on the other hand appear to be helpful. The area under the curve (AUC) for prediction of low-risk candidates was 0.786 and for high-risk candidates 0.796. The addition of troponin in a subset increased the AUC to >0.8 for both.

**Conclusions** A majority of patients with chest pain cared for by the EMS suffer from a low-risk condition and have no prognostic reason for acute hospital care given their diagnosis on hospital discharge. A smaller proportion has a high-risk condition and is in need of prompt specialist care. Building models with good accuracy for prehospital identification of these groups is possible. The use of risk stratification models could make a more personalised care possible with increased patient safety.

## Strengths and limitations of this study

► Unselected inclusion of a close to complete county population of patients with chest pain contacting the emergency medical services.

► Low rates of missing considering the prehospital nature of data.

► Well examined cohort including data on demographics, previous medical history, symptoms, vital signs, ECG, biochemical markers and diagnosis on discharge.

► Some variables are entailed with high rates of missing when compared to studies conducted in the hospital setting.

► Study conducted in one county reduces generalisability.

project (the BRIAN (BRöstsmärta I Ambu-laNs (swedish), EMS Chest pain (english)) research programme) on improved prehospital risk assessment of patients with chest pain.

Chest pain is one of the most common chief complaints when contacting the emergency medical services (EMS). About 10%–15% of all EMS missions concern patients with chest pain.[1 2] The role of the EMS when caring for these patients has changed since the introduction of telemedicine solutions for transmission of ECG and direct transport to percutaneous intervention (PCI) centres for patients with ST-elevation myocardial infarction (STEMI).[3] However, these cases constitute only three percent of all patients with chest pain and one third of all patients with acute myocardial infarction (AMI).[2]

Previous studies[4] report that 15% of all EMS patients with chest pain have a life-threatening condition. It has been suggested that a large proportion of the patients transported by the EMS due to chest pain could be cared for outside the hospital.[5] Thus we

## BACKGROUND

This report is the first presenting quantitative results in a larger prospective research

can conclude that patients seen by the EMS before arrival in hospital due to chest pain form a mixed group with different care needs. Guidelines have focused only to a limited extent on personalising the care of patients with chest pain of other origins than STEMI despite the fact that the proportion of patients with AMI has decreased in recent decades from 30%[6] of all patients seen by the EMS due to chest pain to 10%.[2 4]

A more differentiated and personalised care for these chest pain patients has the potential of improving efficiency and outcome. For example, patients with non-STEMI may benefit from bypassing hospitals without PCI capabilities and bypassing the emergency department (ED) for direct transport to cardiac care units (CCU).[7–10] Furthermore, the undifferentiated care of the large group of patients with chest pain contributes to the problematic crowding of EDs.[11 12] Waiting in the ED increases the risk of an adverse outcome for these patients.[13 14] Consistently transporting patients with low-risk conditions to the ED also entails large costs[15] and occupy limited EMS resources. Referral to other destinations than the ED or non-conveyance for low-risk patients could provide improvements in all these aspects.

Hospital research on risk stratification is extensive but research based on data acquired in the prehospital setting is limited.[16] It is important to base prehospital clinical practice on prehospital data since patients who are seen in the prehospital setting differ from those seen at the ED.[6 17–19] This difference might affect pretest probabilities hence diminishing the validity of the models that have been created in the ED[20–22] when applied in a new (prehospital) setting.

Previous research shows that prehospital risk assessment of patients with chest pain may be possible.[2 16 23 24] However, research is still sparse and often has methodological shortcomings, mainly regarding patient selection and using AMI as the primary endpoint.[20–22] This results in the neglect of other high-risk conditions, such as pulmonary embolism or aortic dissection, and reduces the possibility to identify low-risk patients suitable for non-conveyance. A risk assessment tool for chest pain patients, to be used in the prehospital setting, has been called for in previous research.[25–27] Such a tool might reduce arbitrariness in patient assessment and excess utilisation of emergency care.

The present report is mainly descriptive and investigates contemporary patient characteristics, outcome and how outcome can be predicted by information accessible in the EMS context. Future reports within this project will focus on how such information could be used and combined in a refined risk assessment tool to be used in the prehospital setting in order to stratify risks and refer patients with chest pain to an appropriate level of care.

## Objectives
► To describe contemporary characteristics and diagnoses among prehospital patients with high-risk/low-risk conditions presenting with chest pain.

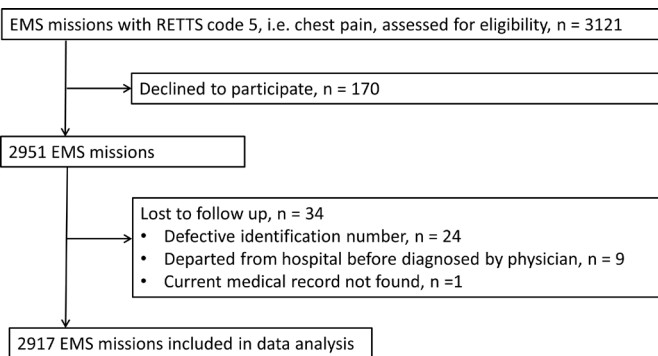

**Figure 1** Flow chart of inclusion process. EMS, emergency medical service; RETTS, Rapid Triage and Treatment System.

► To identify factors suitable for the early recognition of:
– Patients with time-sensitive conditions in need of immediate care (high-risk conditions).
– Patients with no medical need of hospital treatment, suitable for non-conveyance to hospital (low-risk conditions).
– Present data that can inform the development of a prediction tool.

## METHODS
### Study population
This prospective cohort study was conducted in the county of Halland, Sweden. In all, 3121 EMS missions were carried out in 2018 concerning patients, ≥18 years old, with chest pain. All these missions were eligible for inclusion. After excluding patients declining to participate and patients who were lost to follow-up, 2917 EMS missions were included in data analyses (figure 1).

### Healthcare system
The county of Halland covers an area of 5500 km$^2$ and had 329 000 inhabitants in 2018. These are served by two emergency hospitals, including one with PCI capabilities. The EMS consists of 8 ambulance stations with 19 ambulance vehicles. In 2018, a total of 30 672 missions were carried out by the EMS (interhospital site transports excluded). The EMS is staffed mainly by nurses. An ECG is routinely collected prehospitally and stored digitally using telecommunications.

### Data collection
The unique personal identity number assigned to all inhabitants in Sweden made it possible to track each patient throughout the healthcare chain, from EMS mission to hospital discharge and beyond. A novel questionnaire containing fifteen items concerning patient symptoms (online supplemental file 1) was integrated in the digital journal system used by the EMS. The questionnaire was developed by the research group using the findings from two earlier reports within the research project[16 28] along with results and methodology of numerous other studies concerning patients with chest pain. By using electronic

tablets, both the questionnaire and the EMS record were available at the bedside during the entire EMS mission.

The questionnaire (online supplemental file 1) contained items mainly focusing on the patients' pain narratives identifying onset, provocation/palliation, quality, radiation and severity. Intensity was measured using the Numerical Rating Scale (NRS) ranging from 0 to 10.[29] If the patient was not able to use the NRS, a Verbal Rating Scale (VRS) ranging from 'no pain' to 'unbearable pain' was used instead.[30] The answers from the VRS were then transformed into NRS values as follows: 'no pain'=0, 'mild pain'=2 'moderate pain'=5, 'severe pain=7' and 'unbearable pain'=9.[31] The questionnaire also contained items regarding nausea/vomiting, dyspnoea, paleness and clamminess.

From the EMS record, data regarding first measured vital signs were collected. From the hospital and primary care medical records, data on diagnosis on discharge from hospital and previous medical history were retrieved.

During the EMS mission, a blood sample was obtained and brought to the ED. This blood sample was analysed in hospital for high-sensitive troponin T using Roche Cobas e 601, detecting values ≥5 ng/L, cut-off was set to >14 ng/L. Retrieved high-sensitive troponin T-values were also converted into the following intervals <50, 51–100, 101–1000 and >1000 ng/L. In this way, we could stipulate data on troponin T as it would have been presented if Roche's device for bedside troponin T analysis, Cobas h 232, had been used by the EMS.

All registered ECGs were interpreted by either KW or SA using a preset ECG interpretation template (online supplemental file 2). Both were blinded to all patient data except age and sex. In cases of uncertainty, interpretation was discussed with senior research cardiologist ML and JH until consensus was reached. Two hundred ECGs were interpreted separately by both KW and SA concerning identification of ST-elevation, ST-depression, right or left bundle branch block. The kappa coefficient for agreement was 0.810, implying almost perfect agreement.[32] The specific ECG abnormalities included in the kappa coefficient calculation were chosen based on being most clinically relevant for identification of myocardial ischaemia according to the European Society of Cardiology.[33 34]

All data except ECG and diagnosis on discharge from hospital were extracted automatically using digital software. Diagnosis on discharge from hospital, according to International Statistical Classification of Diseases and Related Health Problems 10 (ICD-10), was collected manually from the medical record by KW. Automatically, retrieved data were checked extensively against the original data sources without finding any discrepancies.

### Sample size

The planned sample size was 1500 EMS missions. For a factor present at 10% of all observations, this would entail that a relative difference of 64% and an absolute difference of 9% would be statistically detected. For a factor present at 20% of all observations, this would entail that a relative difference of 49% and an absolute difference of 7% would be detected. For these calculations, 80% power, significance level of 5% (two sided) and a 15% incidence rate of high-risk conditions are applied. Sample size was later increased to offset the higher rates of missing than predicted in respect of specific variables that were seen early in the inclusion process.

### Endpoint

The primary endpoint was a risk-classification group, in terms of low-risk or high-risk condition. All patients included were classified as having either a low-risk, intermediate-risk or high-risk condition as the cause of their chest pain. The adjudication was based on diagnosis on discharge from hospital according to the physician in charge. A high-risk condition was defined as a time-sensitive condition with an increased risk of death and in need of immediate care from a medical point of view in hindsight, for which transport to hospital with highest priority was called on. An intermediate-risk condition was defined as a diagnosis probably in need of hospital care, but for which time was not judged as a critical factor. A low-risk condition was defined as a diagnosis with no medical need of hospital treatment, suitable for non-conveyance to hospital. This group also included patients with chest pain who remained at home and did not visit the ED within 72 hours and who did not die within 30 days. The risk classification of each diagnosis was carried out independently by KW, JH, ML and AB using a preset definition for each risk group blinded to patient characteristics. Thereafter, differences that had arisen were discussed until consensus was reached. Risk classification was also carried out by an external cardiologist. When comparing this external risk classification with that of the article authors, a kappa coefficient of 0.631, substantial agreement, was reached.[32]

### Statistical analysis

All variables were analysed using univariate logistic regression. Each variable was analysed twice, once to test association with high-risk conditions and once to test association with low-risk conditions. If a variable was present in fewer than six patients in any of the groups compared, no analysis was carried out. P values below 0.05 were considered statistically significant. Variables with a statistically significant association were then included in a forward stepwise logistic regression to retain the final independent predictive variables for high-risk, respectively, low-risk conditions. These multivariate analyses were executed on a data set of 714 complete cases. Patients with ST-elevation on ECG were excluded given that there are already well-established fast-tracks for these patients. Tnt were excluded from the multivariate analyses due to high rates of missing. Additional multivariate analyses where Tnt was included were also conducted. However, the study was not powered for these specific analyses and the results should therefore be interpreted with caution. All analyses were carried out using IBM SPSS Statistics V.26.

## Patient and public involvement

Patients have not been directly involved in planning or conducting this study. Design of the questionnaire (online supplemental file 1) was partly based on patient narratives from a previous study within this research project[28] and other studies based on patient interviews. Furthermore, KW had personal contact with several patients contacting him by phone or email due to the opt-out procedure, both patients wanting to opt-out and those who wanted to remain within the study. Results of the study will presented directly to those patients who requested this when contacting KW.

## RESULTS

In total, 139 patients representing 170 EMS missions declined to participate. These patients represent 6% of eligible patients or 5% of eligible EMS missions. In total, 2917 EMS missions concerning 2352 unique patients with non-traumatic chest pain were included in the data analysis. These constitute 10% of all patient-related EMS missions during this time period (interhospital transports excluded). Priority 1 (highest priority) was given by the emergency medical dispatcher to 63% of all missions included, and these constitute 11% of all priority 1 missions within the region. The EMS nurse then triaged patients on scene using the Rapid Triage and Treatment System (RETTS)[35 36] to the highest priority in 10% of cases and to the lowest priority in four percent. EMS missions were equally distributed regarding sex. Median age was 72 years (Q25–Q75, 58–82) (online supplemental file 3).

In total, 16% were classified with a high-risk condition, 16% had an intermediate-risk condition and 68% had a low-risk condition (table 1). The all-cause mortality rate within 30 days was 2.9% (each unique patient counted only once). In the high-risk group, mortality rate within 30 days was 8.9% and in the low-risk group 0.5%.

The cohort was diagnosed with a wide range of medical conditions involving most of the body's organ systems. Most common was musculoskeletal or unspecified chest pain. This diagnosis was given in 41.5% of all EMS missions. Twelve per cent were given an AMI diagnosis on discharge from hospital. In almost ten percent of the EMS missions, the patient was not conveyed to the ED and did not have a related ED visit within 72 hours or died within 30 days (table 1).

Patient characteristics and patient presentation differ substantially within the cohort. However, there are some dominating aspects. Patients are old and have a substantial comorbidity—most commonly hypertension or mental disorders. They commonly describe central, constant, pressuring pain about the size of a palm starting several hours earlier during rest. Vital signs are most commonly unaffected. In most cases, the ECG presents alterations known to be associated with underlying pathological conditions. These are the most common findings in patients with both low-risk and high-risk conditions (online supplemental files 4 and 5).

## Risk prediction

In total, 26 variables showed significantly increased odds and 16 variables showed significantly decreased OR of having a high-risk condition in the univariate analyses (online supplemental file 4). When predicting low-risk conditions, 11 variables showed significantly increased OR for a low-risk condition whereas 37 variables showed a significant increase in the ability to predict the absence of a low-risk condition in the univariate analyses (online supplemental file 5).

When including these variables in a complete cases multivariate analyses 13 and 14 variables remained for high-risk and low-risk prediction, respectively. The predictive variables were largely common to both low-risk and high-risk prediction, with old age and ST-depression as the most important predictors. Medical history, except psychiatric disorders, atrial fibrillation/flutter and chronic obstructive pulmonary disease, had no predictive value in neither high-risk or low-risk models. In total high-risk and low-risk prediction models shared six predictive variables, which are written in italics in table 2. These variables were; age, premature atrial contractions, ST-depression, pain in right arm, paleness and previous medical history of atrial fibrillation/flutter. The area under the receiving operating characteristic curve for prediction of low-risk conditions was 0.786 and 0.796 for high-risk conditions (figure 2). The additional analyses where Tnt were included resulted in a high-risk model with an AUC of 0.847 and a low-risk model with an AUC of 0.884. Accuracy, in terms of AUC, when adding Tnt thereby increased by 8% for high-risk prediction and 11% for low-risk prediction.

## DISCUSSION

To our best knowledge, this is the first prospective population wide study of unselected prehospital chest pain patients beyond acute coronary syndromes. This is important since the rule out of AMI alone does not allow EMS to alter conveyance. We present models that can identify high-risk and low-risk underlying conditions with high accuracy and show that most of these patients suffered from a low-risk condition while 16% had an underlying condition with high risk, such as AMI, pulmonary embolism, aortic dissection, arrhythmias, sepsis and bleeding gastric ulcer. This variety of diagnoses stresses the incentive to have broad inclusion criteria in contrast with previous reports.

Accuracy in terms of area under the curve (AUC) and recieving operating characterstics (ROC) curve shape is quite similar when comparing low-risk and high-risk prediction models. Both models also seem to improve substantially if adding Tnt.

The present paper aims to inform future risk assessment tools with capabilities to rule in high-risk conditions and rule out low-risk separately. To that end and opposite to expectations classic symptoms of AMI like radiation to left arm or jaws, constant pain, clamminess and dyspnoea

**Table 1** Distribution of risk classification and diagnoses by sex and age

| | All % (n) | Sex | | Age group | | |
| --- | --- | --- | --- | --- | --- | --- |
| | | Men % (n) | Women % (n) | ≤50% (n) | 51%–64% (n) | ≥65% (n) |
| | 100 (2917) | 50.2 (1465) | 49.8 (1452) | 16.3 (476) | 18.5 (539) | 65.2 (1902) |
| **High-risk conditions** | 16.0 (467) | 20.3 (298) | 11.6 (169) | 6.1 (29) | 14.8 (80) | 18.8 (358) |
| NSTEMI | 6.7 (194) | 8.2 (120) | 5.1 (74) | 2.1 (10) | 5.0 (27) | 8.3 (157) |
| STEMI | 4.3 (127) | 5.9 (87) | 2.8 (40) | 1.3 (6) | 6.5 (35) | 4.5 (86) |
| Unstable angina pectoris | 2.1 (60) | 2.8 (41) | 1.3 (19) | 0.4 (2) | 1.5 (8) | 2.6 (50) |
| Pulmonary embolism | 0.8 (24) | 1.0 (14) | 0.7 (10) | 0.8 (4) | 0.7 (4) | 0.8 (16) |
| Undefined AMI* | 0.4 (11) | 0.7 (10) | 0.1 (1) | 0.0 (0) | 0.2 (1) | 0.5 (10) |
| MINCA/MINOCA | 0.2 (7) | 0.1 (1) | 0.4 (6) | 0.0 (0) | 0.9 (5) | 0.1 (2) |
| Aortic dissection/aneurysm | 0.2 (7) | 0.4 (6) | 0.1 (1) | 0.0 (0) | 0.0 (0) | 0.4 (7) |
| Severe arrhythmias and conducting disorders† | 0.2 (7) | 0.3 (5) | 0.1 (2) | 0.2 (1) | 0.0 (0) | 0.3 (6) |
| Takotsubo | 0.2 (5) | 0.1 (1) | 0.3 (4) | 0.4 (2) | 0.0 (0) | 0.2 (3) |
| Sepsis | 0.1 (4) | 0.2 (3) | 0.1 (1) | 0.2 (1) | 0.0 (0) | 0.2 (3) |
| Stroke/TIA | 0.1 (4) | 0.1 (2) | 0.1 (2) | 0.4 (2) | 0.0 (0) | 0.1 (2) |
| Gastric ulcer with perforation/bleeding | 0.1 (4) | 0.1 (1) | 0.2 (3) | 0.0 (0) | 0.0 (0) | 0.2 (4) |
| Pulmonary oedema | 0.1 (2) | 0.0 (0) | 0.1 (2) | 0.0 (0) | 0.0 (0) | 0.1 (2) |
| Other, high risk | 0.4 (11) | 0.5 (7) | 40.3 | 0.2 (1) | 0.0 (0) | 0.5 (10) |
| **Intermediate-risk conditions** | 15.6 (455) | 16.7 (245) | 14.5 (210) | 8.6 (41) | 10.8 (58) | 18.7 (356) |
| Atrial fibrillation/flutter | 3.8 (112) | 3.9 (57) | 3.8 (55) | 0.6 (3) | 3.0 (16) | 4.9 (93) |
| Heart failure (without pulmonary oedema) | 2.1 (60) | 2.3 (34) | 1.8 (26) | 0.0 (0) | 0.2 (1) | 3.1 (59) |
| Pneumonia | 1.8 (52) | 1.7 (25) | 1.9 (27) | 0.2 (1) | 0.7 (4) | 2.5 (47) |
| Myocarditis, pericarditis, endocarditis | 1.1 (31) | 1.6 (23) | 0.6 (8) | 2.5 (12) | 1.3 (7) | 0.6 (12) |
| Syncope and collapse | 0.6 (18) | 0.5 (7) | 0.8 (11) | 0.4 (2) | 0.7 (4) | 0.6 (12) |
| Aortic valve stenosis | 0.6 (17) | 0.7 (10) | 0.5 (7) | 0.0 (0) | 0.2 (1) | 0.8 (16) |
| Tumour | 0.6 (17) | 0.5 (7) | 0.7 (10) | 0.2 (1) | 0.4 (2) | 0.7 (14) |
| Chronic obstructive pulmonary disease | 0.5 (14) | 0.6 (9) | 0.3 (5) | 0.0 (0) | 0.0 (0) | 0.7 (14) |
| Infection, intermediate risk | 0.5 (14) | 0.5 (7) | 0.5 (7) | 0.4 (2) | 0.6 (3) | 0.5 (9) |
| Supraventricular tachycardia | 0.4 (13) | 0.3 (5) | 0.1 (2) | 0.6 (3) | 0.7 (4) | 0.3 (6) |
| Cholelithiasis | 0.4 (13) | 0.5 (7) | 0.4 (6) | 0.4 (2) | 0.4 (2) | 0.5 (9) |
| Pancreatitis | 0.3 (9) | 0.3 (5) | 0.3 (4) | 0.6 (3) | 0.2 (1) | 0.3 (5) |
| Electrolyte disturbance | 0.2 (7) | 0.5 (7) | 0.0 (0) | 0.2 (1) | 0.0 (0) | 0.3 (6) |
| Convulsions and seizures | 0.2 (6) | 0.3 (4) | 0.1 (2) | 0.6 (3) | 0.4 (2) | 0.1 (1) |
| Diverticulitis | 0.1 (4) | 0.3 (4) | 0.0 (0) | 0.0 (0) | 0.0 (0) | 0.2 (4) |
| Other, intermediate risk | 2.3 (67) | 2.5 (37) | 2.1 (30) | 1.7 (8) | 2.0 (11) | 2.5 (48) |
| **Low-risk conditions** | 68.4 (1995) | 62.9 (922) | 73.9 (1073) | 85.3 (406) | 74.4 (401) | 62.5 (1188) |
| Chest pain, unspecified | 41.5 (1211) | 39.0 (572) | 44.0 (639) | 52.7 (251) | 50.8 (274) | 36.1 (686) |
| Did not convey (no related ED visit in 72 hours or death within 30 days) | 9.5 (276) | 8.1 (118) | 0.3 (158) | 16.4 (78) | 8.7 (47) | 7.9 (151) |
| Angina pectoris (unstable and spasm induced angina excluded) | 3.1 (90) | 3.3 (48) | 2.9 (42) | 0.4 (2) | 1.5 (8) | 4.2 (80) |
| Abdominal and pelvic pain | 2.1 (62) | 1.4 (21) | 41 (2.8) | 2.7 (13) | 1.7 (9) | 2.1 (40) |

Continued

| | | Sex | | Age group | | |
|---|---|---|---|---|---|---|
| | All % (n) | Men % (n) | Women % (n) | ≤50% (n) | 51%–64% (n) | ≥65% (n) |
| Infection, low risk | 1.6 (47) | 1.3 (19) | 1.9 (28) | 1.1 (5) | 1.9 (10) | 1.7 (32) |
| Gastritis/gastro-oesophageal reflux disease | 1.4 (41) | 1.2 (18) | 1.6 (23) | 2.1 (10) | 1.3 (7) | 1.3 (24) |
| Dyspnoea and coughing | 1.4 (40) | 1.5 (22) | 1.2 (18) | 1.7 (8) | 1.1 (6) | 1.4 (26) |
| Palpitation and benign arrhythmias | 1.2 (35) | 0.8 (11) | 1.7 (24) | 1.7 (8) | 1.1 (6) | 1.1 (21) |
| Anxiety and other mental disorders | 0.9 (25) | 0.6 (9) | 1.1 (16) | 2.7 (13) | 0.2 (1) | 0.6 (11) |
| Other pain | 0.7 (19) | 0.6 (9) | 0.7 (10) | 0.2 (1) | 1.3 (7) | 0.6 (11) |
| Anaemia | 0.5 (16) | 0.5 (8) | 0.6 (8) | 0.2 (1) | 0.7 (4) | 0.6 (11) |
| Vertigo | 0.5 (14) | 0.5 (7) | 0.5 (7) | 0.4 (2) | 0.2 (1) | 0.6 (11) |
| Back pain | 0.3 (10) | 0.2 (3) | 0.5 (7) | 0.2 (1) | 0.2 (1) | 0.4 (8) |
| Orthostatic hypotension | 0.3 (10) | 0.5 (8) | 0.1 (2) | 0.0 (0) | 0.0 (0) | 0.5 (10) |
| Hypertension | 0.3 (10) | 0.1 (2) | 0.6 (8) | 0.2 (1) | 0.2 (1) | 0.4 (8) |
| Mental and behavioural disorders due to alcohol | 0.3 (10) | 0.6 (9) | 0.1 (1) | 0.4 (2) | 0.7 (4) | 0.2 (4) |
| Headache | 0.2 (6) | 0.1 (1) | 0.3 (5) | 0.6 (3) | 0.2 (1) | 0.1 (2) |
| Other, low risk | 2.5 (74) | 2.6 (38) | 2.5 (36) | 1.5 (7) | 2.6 (14) | 2.8 (53) |

*Type of myocardial infarction not stated in patient medical record.
†Ventricular tachycardia, AV-block III.
AMI, acute myocardial infarction; ED, emergency department; MINCA, myocardial infarction with normal coronary arteries; MINOCA, myocardial infarction with non-obstructive coronary arteries; NSTEMI, non-ST-elevation myocardial infarction; STEMI, ST-elevation myocardial infarction; TIA, transient ischemic attack.

did not differ between high- and low-risk conditions in the multivariate analyses. The same was true for previously known risk factors such as history of hypertension, ACS and diabetes.[34] These findings are surprising. It may be explained by our study using a wider range of diagnoses as endpoint compared with previous studies focusing on AMI.[5 37] This is strengthened by ad hoc analyses showing increased significance for above-mentioned factors when using AMI as endpoint instead of high-risk condition. It may also be explained by prehospital patients having a higher comorbidity compared with patients at the ED[6] and that medical history, therefore, is less useful to discriminate high-risk from low-risk conditions in the prehospital setting. However, most of these factors differ in the univariate analyses but not in the multivariate ones. Thereby, these factors may still have a predictive value, but when combined with others predictive factors they become less important. Future risk models may need to cover other variables, with ability to contribute to accuracy, than traditional risk factors for AMI. The use of artificial intelligence may be one way to identify such variables.

In both the low-risk and high-risk models, premature atrial complex (PAC) on ECG have a predictive value. This finding was not expected. Once again, this may be explained by not using AMI as endpoint. This is strengthened by ad hoc analyses using AMI as endpoint where PAC did not turn out significantly. Given the multiple

analyses and the quite wideCIs, this may also be a chance finding.

Vital signs seem to have little predictive value when identifying high-risk conditions. This finding is somewhat surprising, however not unique. For example, Frisch et al[38] reported no association between prehospital heart rate/blood pressure and need of 'advanced hospital cardiac care'. To some extent, this may be explained by strongly deviating vital signs were quite rare within the cohort, that is, the vast majority of patients with chest present with normal vital signs. However, tachypnoea, low oxygen saturation and fever all reduced the odds for a low-risk condition. Deviating vital signs thereby seem more useful when ruling out patients from the low-risk group rather than to predict high-risk conditions.

One main finding of this study, and maybe also most relevant in clinical care, is that more than two thirds of all EMS missions concerning patients with chest pain were classified as low-risk, that is, having no medical need for acute transportation to hospital given the diagnosis on discharge. This implies the substantial magnitude of the impact of a predictive model identifying them with high accuracy early in the care chain. Diverting only a proportion of these low-risk patients away from the ED to less resource intensive venues would probably reduce healthcare costs, EMS workload and ED crowding. However, one must have in mind that such a predictive model only refers to the strict medical aspects of being in need of

**Table 2** Predictors of high-risk and low-risk conditions after multivariate analysis

| | OR | 95 % CI | P value* |
|---|---|---|---|
| **High-risk predictors** | | | |
| *Age ≤50* | – | – | <0.001 |
| *Age 51–64* | 18.98 | 5.47 to 65.95 | 0.000 |
| *Age ≥65* | 7.87 | 2.12 to 29.24 | 0.002 |
| *ECG premature atrial contractions (PAC)* | 4.52 | 1.67 to 12.25 | 0.003 |
| *ECG ST-depression* | 3.52 | 1.63 to 7.58 | 0.001 |
| *Pain in right arm* | 2.80 | 1.48 to 5.32 | 0.001 |
| *Paleness* | 2.44 | 1.44 to 4.14 | <0.001 |
| Male | 2.01 | 1.29 to 3.14 | 0.002 |
| Debut during activity | 2.01 | 1.22 to 3.32 | 0.006 |
| Pain intensity according to Numeric Rating Scale >8 | 2.38 | 1.10 to 5.13 | 0.026 |
| ECG T-wave inversion | 2.01 | 1.06 to 3.82 | 0.032 |
| Time elapsed since pain onset >3 hours | 0.49 | 0.31 to 0.77 | 0.001 |
| *Previous atrial fibrillation/flutter* | 0.42 | 0.25 to 0.72 | 0.001 |
| Previous COPD | 0.41 | 0.18 to 0.94 | 0.035 |
| Size of area affected by pain; two inch diameter | 0.35 | 0.13 to 0.93 | 0.035 |
| **Low-risk predictors** | | | |
| Previous atrial fibrillation/flutter | 2.46 | 1.47 to 4.11 | <0.001 |
| Right sided chest pain | 3.88 | 1.30 to 11.59 | 0.015 |
| Previous psychiatric diagnosis (any type) | 1.70 | 1.14 to 2.54 | 0.009 |
| Sinus rhythm without abnormalities | 1.70 | 1.11 to 2.60 | 0.014 |
| Paleness | 0.39 | 0.25 to 0.63 | <0.001 |
| *Breathing rate ≥25 breaths/min* | 0.45 | 0.24 to 0.82 | 0.009 |
| ECG atrial fibrillation/flutter | 0.42 | 0.23 to 0.76 | 0.004 |
| *Age ≤50* | – | – | <0.001 |
| *Age 51–64* | 0.40 | 0.18 to 0.87 | 0.021 |
| *Age ≥65* | 0.22 | 0.11 to 0.45 | <0.001 |
| *Pain in right arm* | 0.28 | 0.15 to 0.50 | <0.001 |
| Pain in right shoulder | 0.33 | 0.14 to 0.80 | 0.014 |
| *ECG PAC* | 0.36 | 0.14 to 0.94 | 0.036 |
| Oxygen saturation ≤91 % | 0.32 | 0.12 to 0.84 | 0.020 |
| *ECG ST-depression* | 0.18 | 0.08 to 0.42 | <0.001 |
| Body temperature >38.0°C | 0.15 | 0.04 to 0.63 | 0.009 |

*Italics*=Variable common for both high-risk and low-risk condition prediction models.
*Stepwise forward logistic regression, complete cases, entery p<0.05, removal p<0.10
COPD, chronic obstructive pulmonary disease.

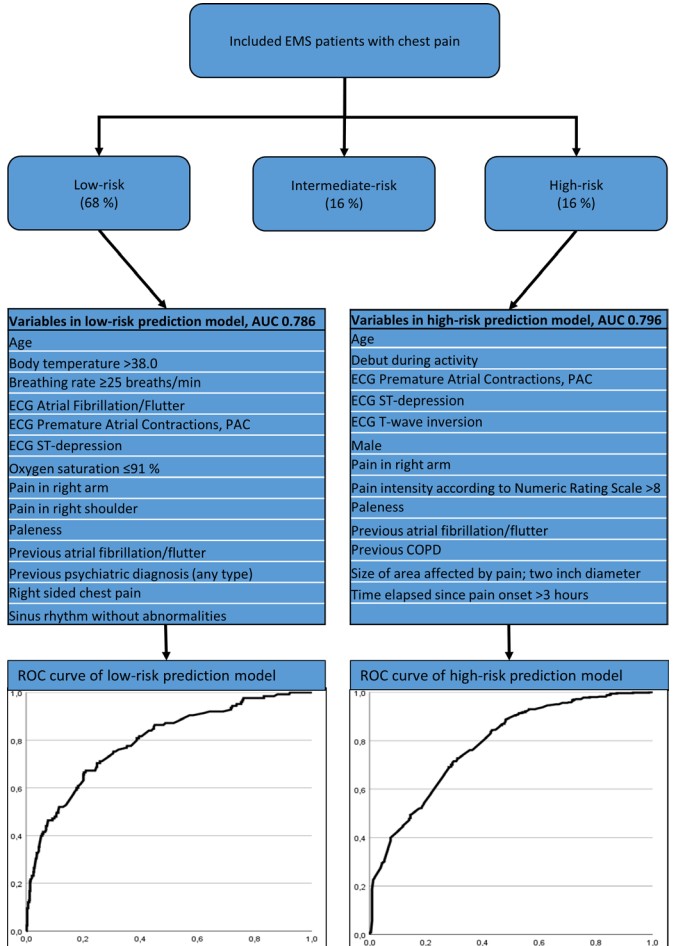

**Figure 2** Schematic presentation of prediction models accuracy. AUC, area under the curve; COPD, chronic obstructive pulmonary disease; EMS, emergency medical service; ROC, recieving operating characterstics.

EMS personnel but other factors than medical risk status must always be considered when assessing patient care needs. It is also of importance to include the patient in the decision making process and take the patient perspective into account.

Among chest pain patients, ten percent remained at home, despite EMS personnel having neither mandate nor instruments to provide such recommendations. This finding indicates that even today, EMS personnel or the patients themselves sometimes deem transport to the ED as inappropriate, despite the patients' chest pain. However, little is known about the appropriateness of EMS decision making when suggesting this type of triage.

Among the 16% of EMS missions with patients with high-risk conditions, ACS constituted the major part. The portion of high-risk conditions is in line with previous findings.[2 4] Importantly STEMI constitute less than one-third of all missions concerning high-risk conditions, confirming work by Pedersen *et al*.[2] The fairly low incidence of STEMI shows the importance of taking other high-risk diagnoses into account when risk-assessing patients with chest pain. This would for example enable direct transport to CCU, hospitals with PCI laboratory

acute hospital care. There can be other circumstances that promote transportation to hospital by ambulance. For example, intense anxiety, patient being unable to handle the situation, comorbidities such as dementia. Such a prediction model should be used to guide the

or capabilities for thoracic surgery.[10] However, referring patients to such instances should be done with great caution given the inclusion of several non-cardiac diagnoses in the high-risk group. A refined model focusing on identification of cardiac patients could be a feasible way to improve the possibilities for such direct transportation patient management.

Seeking to discriminate between high-risk and low-risk conditions, we also looked for information that would not support this aim. In this report, we; therefore, acknowledge that symptoms typically associated with myocardial ischaemia, such as central pain, pressuring pain quality, left arm radiation and affected breathing, are common in both groups. Notably most patients with such symptoms were found in the low risk group. This may limit the usefulness of some symptoms previously reported to be of value in risk assessment and triage. However, other aspects of symptoms such as pain intensity, pain in the right arm and the presence of paleness were helpful in the discrimination of high versus low risk.

Risk assessment in the prehospital setting may serve several purposes. Conveyance decision based on the possibility to identify both low-risk patients suitable for non-hospital care/non-conveyance and high-risk patients in need of prompt specialist care are separate issues.

To further inform a future model supporting the conveyance decision a multivariate analysis was reported. The results of the multivariate analyses presented in this study could form the basis for such a decision support tool. The c-statistics indicate that such a tool could reach a level of accuracy appropriate for clinical use. The fact that several classic symptoms and previously known risk factors of AMI turned out to be of little predictive value indicates that previous risk prediction tools developed in the hospital setting such as the History ECG Age Risk Factors Troponin (HEART)-score and RETTS may be less accurate and invalid when used in the prehospital setting. It is also important to acknowledge that a single factor in itself can constitute grounds for prompt transport to hospital, such as strongly deviating vital signs.

The models presented in this study need to be refined before tested further. Adding Tnt to the analyses seems to improve accuracy substantially and needs to be investigated further. Analyses on imputed data or a new data set with more complete data regarding Tnt is needed. It would also be preferable to develop a combined model predicting both low-risk and high-risk conditions, given that good c-statistics may be retained. It would also be of interest to examine the accuracy of reduced models with fewer variables. A combined model including fewer variables would ease clinical usage. The result of such a refined model development will be presented in upcoming reports within the research programme. Prediction models developed using artificial intelligence will also be presented. External validation of the final prediction model(s) is planned to be carried out in multicentre studies if upcoming results indicate that clinical usage would be appropriate. The prediction model(s) could be integrated in a mobile application or other electronic device to improve clinical usability.

## Strengths and limitations

A key strength is the complete picture of the study sample as data provides information on demographics, medical history, a wide range of symptoms, vital signs, ECG, biochemical markers and diagnosis on discharge. However, some variables included entail rather high rates of missing information (online supplemental files 4 and 5). When analysing the distribution of missing data, no skewness of clinical relevance could be identified. This is also confirmed by the fact that the results of the univariate analyses to a large extent confirm previous findings concerning which variables were associated with high-risk conditions. This holds true also for variables with the highest rates of missing, for example, pain intensity,[39] pain radiation,[39] quality of pain[21] and biochemical markers (Tnt).[40]

Data collection in the prehospital setting is known to be challenging. The situation is often perceived as stressful by the patient, the patient may be physically or mentally unable to provide the information requested, personnel resources are limited and protocol compliance among EMS personnel has been reported to be low.[41 42] All these components contribute to the finding that prehospital research often suffers from high rates of missing information. In this study, we observed that missing rates were highest regarding data where the questionnaire (online supplemental file 1) required the user to write the answer rather than tick a box. The rates of missing data in these cases can thereby probably mainly be explained by limited protocol compliance among EMS personnel.

The high rate of missing data regarding Tnt is a major limitation, since this complicates the inclusion of Tnt in the multivariate analyses. Upcoming reports, using data that are more complete on Tnt, will present more comprehensive prediction models. There are two main reasons for the high rates of missing data on Tnt. First, no blood sample analysis was done if the patient remained at home. Second, a blood sample could not be analysed if the patient was transported directly to a hospital outside the county. However, these reasons for not obtaining data on Tnt are not a real-life problem if using a device for prehospital bedside troponin T analysis.

Considering that this was a study based on prehospital data, the rates of missing information must be regarded as low and therefore data may be looked on as comparatively comprehensive. The varying rates of missing information show possible difficulties using certain variables in a prehospital context. For this reason, it is also important to be careful when selecting variables to be included in a planned decision support tool. A tool requiring data on variables which are difficult to collect in the prehospital setting will reduce personnel compliance and thereby be of less clinical value.

This study use risk-classification group as endpoint. The classification was based on extensive discussions among

experienced colleagues trying to reach a consensus. Risk classification was also carried out by an external cardiologist, with substantial Cohen's kappa agreement. However, there are not absolute distinctions between risk-classification groups. For example, aortic stenosis and acute pancreatitis can present in different ways. Patients with these diagnoses can be in need of prompt hospital care but in other cases, time is of little importance. Abnormal vital signs may be helpful to discern such cases. Furthermore, there may be other reasons (for example frailty) or intensive pain, which may force the EMS personnel to transport a patient with a low-risk condition to hospital. The objective of the predictions models is to support the EMS personnel in their decision making, not to replace clinical judgement.

### Generalisability

The study is strengthened by the nearly complete and unselected inclusion of all EMS missions within the county concerning patients with chest pain during 2018. This increases the generalisability of the results. However, the use of data from only one EMS organisation negatively impacts the external validity. We have no reason to believe that our result is not applicable in other counties in Sweden. The current county includes both urban and rural areas, with a wide range of socioeconomic conditions and ethnical diversity. However, we do not have access to specific data on these factors for included patients which is a limitation given that those factors are reported to affect the prehospital care of patients with chest pain.[43] Generalisation of the results beyond Sweden should be done with care, at least outside the western world, given differences in EMS organisations, care financing and EMS contact behaviour.

The level of competence among EMS personnel and equipment available differ between different EMS organisations and countries. This should be considered when discussing future clinical implications. With a minimum of training, included variables are easy to obtain, at least if using external ECG interpretation via telemedicine solutions. Future prediction models should therefore be possible to use by EMS personnel with different competence.

Some of the included variables are subjective, either from a patient or personnel perspective. For example, what is regarded as pain in right shoulder may differ among patients. Different EMS personnel may assess paleness differently. The use of such variables should therefore be limited. Instead, more objective parameters such as, ECG, previous medical history, age and sex should be promoted in future models to improve reliability.

### CONCLUSIONS

A majority of patients with chest pain cared for by the EMS suffer from a low-risk condition and have no prognostic reason for acute hospital care given their diagnosis on hospital discharge. A smaller proportion has a high-risk condition and is in need of prompt specialist care. Building models with good accuracy for prehospital identification of these groups is possible. Models optimising rule out will differ from models optimising rule in. However, ECG findings and age are cornerstones in both. Importantly, previously known risk factors such as history of acute coronary syndrome, diabetes or hypertension did not have a predictive value in such models. On the other hand, history of psychiatric disorders and atrial fibrillation/flutter are of importance when risk stratifying prehospital patients with chest pain. The use of risk stratification models will make a more personalised care possible with increased patient safety. More research on the value of adding Tnt and how to refine such risk stratification models is needed before clinical testing.

**Author affiliations**
[1]Institute of Health and Care Sciences, Gothenburg University, Sahlgrenska Academy, Goteborg, Sweden
[2]Department of Ambulance and Prehospital Care, Halland County, Halmstad, Sweden
[3]Department of Molecular and Clinical Medicine/Cardiology, Sahlgrenska Academy, Göteborg, Sweden
[4]Department of Development, Halland Hospital, Halland County, Halmstad, Sweden
[5]The Prehospital Research Center Western Sweden, University of Borås, Borås, Sweden
[6]Department of Cardiology, Halland Hospital, Halland County, Halmstad, Sweden

**Acknowledgements** We want to thank the following persons: The EMS personnel of Region Halland for their help with data collection. Thomas Thorsson, Vinh Norberg Nguyen and Sasa Pejicic for data extraction. Anders Holmén and Ulf Strömberg for statistical support.

**Contributors** KW, ML, JH, SA and AB designed the study and planned the data collection. KW, ML, JH and SA interpreted the ECG's. Data analysis was carried out by KW. KW, ML, JH and AB contributed in writing the manuscript. All authors read and approved the final manuscript.

**Funding** This study has been funded by the Department of Ambulance and Prehospital Care, Region Halland and the Scientific Council of Region Halland (HALLAND-209901).

**Disclaimer** The funding bodies had no role in the design, conduct, interpretation or writing of the report on this research.

**Competing interests** None declared.

**Patient consent for publication** Not required.

**Ethics approval** The study was approved by the Regional Ethical Review Board in Lund (Dnr 2017/212). All patients were given the opportunity to withdraw their participation using an opt-out procedure.

**Provenance and peer review** Not commissioned; externally peer reviewed.

**Data availability statement** Data are available on reasonable request. The datasets generated and analysed during the current study are not publicly available due the integrity of patient privacy but are available from the corresponding author on reasonable request and if approved by the Regional Ethical Review Board in Lund.

**ORCID iD**
Kristoffer Wibring http://orcid.org/0000-0002-6910-230X

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
