## [Reviewer comments · BMJ Open]

ARTICLE DETAILS

TITLE (PROVISIONAL)	Prehospital stratification in acute chest pain patient into high- and low-risk by emergency medical service – a prospective cohort study
AUTHORS	Wibring, Kristoffer; Lingman, Markus; Herlitz, Johan; Amin, Sinan; Bang, Angela

VERSION 1 – REVIEW

REVIEWER	Professor Paul Collinson St Georges Hospital
REVIEW RETURNED	22-Oct-2020

GENERAL COMMENTS	The study is essentially a descriptive statistical exercise. A strength of the study is that it has been conducted in the context of first contact of patients with the emergency medical system (EMS) rather than first hospital attendance and it is of interest that 10% of patients were not transferred to hospital. I would make the following observations on the manuscript as currently structured. 1. There has been extensive work on risk stratification although I accept entirely that this has been largely focused on patients arriving at the Emergency Department (ED). Although I appreciate the authors are looking at first contact rather than ED attendance they do not put their data in the context of the existing studies as to the whether or not there is congruence between their own findings and previous publications. Indeed, much of the work on ED attendance risk scoring has been driven by an appreciation risk scoring commenced with patients with acute coronary syndrome and that such scores would be inappropriate in a chest pain population.2. The authors, not unreasonably, include the ECG as part of the risk assessment. An ECG can be performed on-site and interpreted in real time or assessed by telemetry (ideally both). However, the presence of significant ECG abnormalities especially ST segment elevation would be considered as a cause for immediate transfer to a facility for intervention. Although I understand that the objective of this study is to be inclusive, given that the object of the exercise is to identify low-risk individuals it would seem more appropriate to list those factors which would not immediately be considered as part of decision-making. Unless the authors have evidence that ECG changes would be ignored in routine clinical practice (were any patients with ECG changes not admitted to hospital?) they need to consider more realistically how to analyse their data so that those patients who would automatically be transferred are excluded from the analysis and the data reanalysed and reviewed to see whether or not definitive
--

	ECG changes significantly interact with the other variables they have documented. Hence, in a patient without ECG changes, what is important. 3. I find the approach taken to cardiac troponin data somewhat puzzling. The authors justify the stratification on the basis of the current analytical capability of existing point of care testing (POCT). This is unhelpful. It is anticipated that within the next few years high sensitivity troponin measurement will be available by POCT. Indeed, a high sensitivity troponin I POCT method has been published. Since there is already an extensive body of literature which shows that very low troponin levels at first presentation can be used to predict likelihood of subsequent cardiac events and they have high sensitivity troponin data (albeit with a time delay), they should include this within the analysis and look at the decision points as proposed by the European Society of Cardiology for rapid rule out. It is likely that in the EMS scenario the combination of ECG, troponin (measured on site by a high sensitivity POCT method) and selected clinical features would be a powerful predictor. The interest is in decision-making strategies is going forward not to the technology of 5 years in the past. 4. Unfortunately, the tables included in the supplementary data are very confusing and need to be structured to explain exactly what they contain (and ideally presented in landscape format). I was completely unable to comprehend everything in supplementary material 4 from page 31 onwards.
--	--

REVIEWER	Boogers, Mark Leiden Universitair Medisch Centrum, Cardiology
REVIEW RETURNED	25-Nov-2020

GENERAL COMMENTS	Review manuscript: “Prehospital Stratification in Acute Chest Pain Patient into High- and Low Risk by Emergency Medical Service – A Prospective Study” The study is designed to evaluate a timely and relevant topic regarding the prehospital triage of cardiac patients. A well-defined selection in the prehospital scene could potentially reduce unnecessary ambulance rides and interclinical transfers, leading to improved cardiac care. The topic is interesting as the authors have demonstrated that a predictive model based on conventional clinical and ECG parameters is useful for triage of cardiac patients. However, when reading the manuscript, some issues should be clarified: Major Comments: 1. Improvement in prehospital cardiac triage can indeed be gained in patients with chest pain, predominantly in patients with doubts whether the complaint is of cardiac origin. For instance, the majority of the high risk conditions as defined by the authors could be identified using clinical assessment , ECG and vital sign measurement. For these patients (e.g. STEMI), a predictive model is of no additional interest, one should transfer the patient to an intervention centre without further delay. Even some delay could be introduced using extensive questionnaires. So, my comment is that the authors should underline their evaluated predictors could be of more interest in some patient categories (predominantly intermediate risk cases).
---

	2. In the study, a batch of conventional clinical parameters were not predictive for high risk conditions, including cardiac history and cardiovascular risk factors (DM or hypertension). However, this is an interesting finding, as the currently applied predictive models (for instance the HEART or modified HEART score or the Marburg Risk Score) are based on these factors. On the other hand, we have gained more insight in atypical presentations for ACS in the elderly, diabetics or females. So, in this perspective, I was wondering what the potential explanation is for these study findings (conventional risk markers of CHD not predictive, whereas some subjective parameters (paleness) are predictive)? I find it difficult to believe that 'known coronary artery disease' or diabetic disease is not of importance for triage of cardiac patients. 3. In table 2, the authors have reported predictors for high or low-risk conditions. Why is an ECG with PACs predictive (OR 3.8) for high-risk conditions, whereas a VT or a LBBB is not a predictor? Please comment 4. I do not see the blood pressure, tachycardia or tachypnoea as a marker of illness/high risk conditions. I presume these vital basic parameters have been implemented in the analyses. What could be the reason for its low predictive value? 5. Improved triage at the ambulance should be practical. These study has evaluated extensive parameters for cardiac triage. I would suggest to summarize these predictors in future studies into a concise model useful for clinics. One of the strength of the current HEART score is its simplicity. 6. In prehospital setting, we have to deal with a large set of ambulance professionals. For this reason, one would like to have a standardized protocol using predominantly quantitative parameters. The current model is also based on subjective parameters, like 'paleness' of the patient. Could the authors comment on this? Is this model applicable in different ambulance regions? With different level of expertise/knowledge? 7. The authors have defined 3 categories of endpoints: High risk, intermediate and low risk conditions. These conditions were based on time-dependency; in which high risk conditions were defined as conditions with direct indication for medical care, whereas intermediate conditions were time was not that important. When looking into these categories, I was wondering how the authors decided which clinical entity was placed in which category (for instance: aortic stenosis could be of true importance or could be not relevant at all, also non-cardiac pathophysiologic disease could be classified in another way: pancreatitis can also have a lethal course). Please add more information on the classification of the conditions Minor Comment: 1. In table 1: A total AV block is no arrhythmia. Please change this description 2. If possible, a figure that illustrates the main results would be of additional value. 3. What is the difference between: Undefined MI and MINOCA/MINCA? 4. The lay-out of the tables could be improved. It is difficult to interpret the current extensive tables.
--	---

REVIEWER	Camila Caiado Durham University, UK
REVIEW RETURNED	08-Feb-2021

GENERAL COMMENTS	The paper is quite interesting and warrants publication due to the value of the data collected and its presentation. I'm looking forward to seeing how this project progresses. Discussion on generalization: The data collected is limited to a specific area in Sweden. If a risk score is to be developed, there has to be a discussion on if and how the study will expand to ensure the results are transferable to other parts of Sweden and beyond. Population: I'd like to see a discussion on the socio-economical factors for the target population. What is the ethnic diversity in the target population and the observed cohort? Can other characteristics such as height and weight be included in this paper? If not, would it be possible to include in future publications? Response time: It would be interesting to know what the response time for these events were. What is the expected response time for EMS in this area? Were there any outliers? Pre-existing conditions: When looking at the risk factors considered, there is no clear distinction on which of those were diagnosed previously and/or are long-term conditions for the patient. would it be possible to reorganize the table to highlight known pre-existing conditions vs conditions diagnosed at the time? For a future publication, it would also useful to know when diagnosis for pre-existing conditions took place (e.g. less than 1 year, 1-5 years, etc.) Missing data: I can't find any discussion on missing data. For a dataset of this size, there must have been a small proportion of missingness. Could the authors please add a statement to the paper? If no data was missing, great, please say so. If data was missing, please explain how it was addressed and whether any imputation techniques were used. AUC/ROC: There is a quick mention of the area under the curve for two fo the risk groups with fairly similar AUCs. Would it be possible to present ROCs and/or expand the discussion in this area?
---

VERSION 1 – AUTHOR RESPONSE

Reviewer: 1

Dr. Paul Collinson, St Georges Hospital

Comments to the Author:

The study is essentially a descriptive statistical exercise. A strength of the study is that it has been conducted in the context of first contact of patients with the emergency medical system (EMS) rather than first hospital attendance and it is of interest that 10% of patients were not transferred to hospital. I would make the following observations on the manuscript as currently structured.

1. There has been extensive work on risk stratification although I accept entirely that this has been largely focused on patients arriving at the Emergency Department (ED). Although I appreciate the authors are looking at first contact rather than ED attendance they do not put their data in the context of the existing studies as to the whether or not there is congruence between their own findings and previous publications. Indeed, much of the work on ED attendance risk scoring has been driven by an

appreciation risk scoring commenced with patients with acute coronary syndrome and that such scores would be inappropriate in a chest pain population.

- We have expanded our discussion upon this and added a few references in order to put our data in the context of previous studies. Please, see the revised manuscript.

2. The authors, not unreasonably, include the ECG as part of the risk assessment. An ECG can be performed on-site and interpreted in real time or assessed by telemetry (ideally both). However, the presence of significant ECG abnormalities especially ST segment elevation would be considered as a cause for immediate transfer to a facility for intervention. Although I understand that the objective of this study is to be inclusive, given that the object of the exercise is to identify low-risk individuals it would seem more appropriate to list those factors which would not immediately be considered as part of decision-making. Unless the authors have evidence that ECG changes would be ignored in routine clinical practice (were any patients with ECG changes not admitted to hospital?) they need to consider more realistically how to analyse their data so that those patients who would automatically be transferred are excluded from the analysis and the data reanalysed and reviewed to see whether or not definitive ECG changes significantly interact with the other variables they have documented. Hence, in a patient without ECG changes, what is important.

- Agree! Patients with ST-elevation have now been excluded from the multivariate analyses given that there are already well established fast-tracks for these patients. Please, see the revised manuscript. In the discussion we also entered a paragraph on this subject. Please, see the revised manuscript. However, we find it important to keep the cohort as unselected as possible, since the final aim for the project is to develop a combined prediction model for identification of both low- and high-risk patients.
- Yes, about half of all non-conveyed patients had ECG changes. Most commonly atrial fibrillation/flutter or T-wave inversion but also other abnormalities. To what extent these ECG changes were new or of old age is unknown to us, which is often also the case in the real world setting.

3. I find the approach taken to cardiac troponin data somewhat puzzling. The authors justify the stratification on the basis of the current analytical capability of existing point of care testing (POCT). This is unhelpful. It is anticipated that within the next few years high sensitivity troponin measurement will be available by POCT. Indeed, a high sensitivity troponin I POCT method has been published. Since there is already an extensive body of literature which shows that very low troponin levels at first presentation can be used to predict likelihood of subsequent cardiac events and they have high sensitivity troponin data (albeit with a time delay), they should include this within the analysis and look at the decision points as proposed by the European Society of Cardiology for rapid rule out. It is likely that in the EMS scenario the combination of ECG, troponin (measured on site by a high sensitivity POCT method) and selected clinical features would be a powerful predictor. The interest is in decision-making strategies is going forward not to the technology of 5 years in the past.

- Agree! We have revised our regression analyses concerning Tnt and instead the well-established 14 ng/L is used as cut-off. Please, see the revised manuscript and supplemental material 4 and 5. However, we find it important to describe also what is possible to achieve today using existing POCT-devices for Tnt. Therefore, we have also kept the results of the regression analyses using intervals for Roche's Cobas h 232.
- The results of additional analyses including Tnt has been added. Please, see the revised manuscript. However, these findings must be interpreted with caution since the study was not powered for this specific analysis.
- Upcoming reports based on more complete data, will present more comprehensive prediction models including Tnt. Please, see the revised manuscript.

4. Unfortunately, the tables included in the supplementary data are very confusing and need to be structured to explain exactly what they contain (and ideally presented in landscape format). I was completely unable to comprehend everything in supplementary material 4 from page 31 onwards.

- Layout of supplementary data have been revised.

Reviewer: 2

Dr. Mark Boogers, Leiden Universitair Medisch Centrum

Comments to the Author:

Review manuscript:

“Prehospital Stratification in Acute Chest Pain Patient into High- and Low Risk by Emergency Medical Service – A Prospective Study”

The study is designed to evaluate a timely and relevant topic regarding the prehospital triage of cardiac patients. A well-defined selection in the prehospital scene could potentially reduce unnecessary ambulance rides and interclinical transfers, leading to improved cardiac care. The topic is interesting as the authors have demonstrated that a predictive model based on conventional clinical and ECG parameters is useful for triage of cardiac patients. However, when reading the manuscript, some issues should be clarified:

Major Comments:

1. Improvement in prehospital cardiac triage can indeed be gained in patients with chest pain, predominantly in patients with doubts whether the complaint is of cardiac origin. For instance, the majority of the high risk conditions as defined by the authors could be identified using clinical assessment, ECG and vital sign measurement. For these patients (e.g. STEMI), a predictive model is of no additional interest, one should transfer the patient to an intervention centre without further delay. Even some delay could be introduced using extensive questionnaires. So, my comment is that the authors should underline their evaluated predictors could be of more interest in some patient categories (predominantly intermediate risk cases).

- Agree! Patients with ST-elevation has in the revised version been excluded from the multivariate analyses given that there are already well-established fast-tracks for these patients. Please, see the revised manuscript. We also expanded the discussion on this subject. Please, see the revised manuscript. However, we find it important to keep the cohort as unselected as possible, since the final aim for the project is to develop a combined prediction model for identification of both low- and high-risk patients.

2. In the study, a batch of conventional clinical parameters were not predictive for high risk conditions, including cardiac history and cardiovascular risk factors (DM or hypertension). However, this is an interesting finding, as the currently applied predictive models (for instance the HEART or modified HEART score or the Marburg Risk Score) are based on these factors. On the other hand, we have gained more insight in atypical presentations for ACS in the elderly, diabetics or females. So, in this perspective, I was wondering what the potential explanation is for these study findings (conventional risk markers of CHD not predictive, whereas some subjective parameters (paleness) are predictive)? I find it difficult to believe that 'known coronary artery disease' or diabetic disease is not of importance for triage of cardiac patients.

- We agree! These findings are indeed very interesting and somewhat surprising. We have extended our discussion on this subject presenting plausible explanations. *“It may be explained by our study using a wider range of diagnoses as endpoint compared with previous studies focusing on AMI. This is strengthened by ad-hoc analyses showing increased*

significance for above mentioned factors when using AMI as endpoint instead of high-risk condition. It may also be explained by the fact that prehospital patients conform a relatively sick study cohort having a higher comorbidity compared to patients at the ED and that previous medical history therefore is less useful to discriminate high- from low-risk conditions in this cohort. However, most of the risk factors mentioned above are significant in the univariate analyses but not in the multivariate ones. Thereby, these factors may still have a predictive value, but when combined with other predictive factors they become less important.” Please, see the revised manuscript.

Maybe the prognosis in these patients are improved by protective treatment decreasing their risk as opposed to undiagnosed (hence unprotected) CHD patients in the control group.

3. In table 2, the authors have reported predictors for high or low-risk conditions. Why is an ECG with PACs predictive (OR 3.8) for high-risk conditions, whereas a VT or a LBBB is not a predictor? Please comment

- The predictive value of PAC is a surprising finding indeed.. This may be a chance finding as a result of the multiple analyses. However, it may also be explained by the use of low/high-risk condition as end-point instead of AMI. In ad-hoc analyses using AMI as endpoint PAC did not turn out as a significant predictor. This finding merits further investigation elsewhere. Please, see the revised manuscript where we have added a comment about this.
- Concerning VT this was only present in one of the 2785 ECGs that were interpreted. Therefore we could not perform any statistical analyses including VT. However, in clinical practice VT in itself should be handled as a high-risk condition.
- Indeed, LBBB did not turn out as predictor. The 2020 ESC guidelines for NSTEMI patients state that haemodynamically stable patients presenting with chest pain and LBBB only have a slightly higher risk of having AMI compared to patients without LBBB. Maybe therefore, our result regarding LBBB is not that surprising especially when considering that we used another endpoint than AMI. In addition, in our data like in real world data we cannot distinguish if the LBBB is previously known or not, which will probably reduce the predictive value even more.

4. I do not see the blood pressure, tachycardia or tachypnoea as a marker of illness/high risk conditions. I presume these vital basic parameters have been implemented in the analyses. What could be the reason for its low predictive value?

- The main reason for vital signs did not turn out predictive to any wider extent is probably that strongly deviating vital signs were quite rare. For example a systolic blood pressure < 90 mmHg were only present in 29 of 2917 patients (Supplemental material 5). However, SpO₂, fever and tachypnoea all decreased the risk for low-risk condition (Table 2). From this we conclude that deviating vital signs are more useful to rule-out patients from the low-risk group than to predict high-risk conditions. Please, see the revised manuscript.
- However, we strongly think that deviating vital signs in themselves are reasons enough for prompt hospital care. Please, see the revised manuscript.

5. Improved triage at the ambulance should be practical. These study has evaluated extensive parameters for cardiac triage. I would suggest to summarize these predictors in future studies into a concise model useful for clinics. One of the strength of the current HEART score is its simplicity.

- Agree! We also highlight this in the discussion section. Indeed, the final prediction model must be easy to use if clinical impact should be achieved. Preferably a combined model for both low- and high-risk prediction. We are now planning the next study within this project aiming at refining the models in this manuscript into a combined model with fewer variables but with retained or improved accuracy. Please, see in the revised manuscript.

6. In prehospital setting, we have to deal with a large set of ambulance professionals. For this reason, one would like to have a standardized protocol using predominantly quantitative parameters. The current model is also based on subjective parameters, like 'paleness' of the patient. Could the authors comment on this? Is this model applicable in different ambulance regions? With different level of expertise/knowledge?

- Agree! Parameters that are more objective should be promoted in forthcoming model refinement studies. Please, see the revised manuscript.
- We judge all variables easy to obtain with a minimum level of training, at least if using external ECG interpretation via telemedicine solutions, and should therefore be possible to use by EMS personnel with different competence. Especially if integrating the model into a mobile application or other electronic device. Please, see the revised manuscript.

7. The authors have defined 3 categories of endpoints: High risk, intermediate and low risk conditions. These conditions were based on time-dependency; in which high risk conditions were defined as conditions with direct indication for medical care, whereas intermediate conditions were time was not that important. When looking into these categories, I was wondering how the authors decided which clinical entity was placed in which category (for instance: aortic stenosis could be of true importance or could be not relevant at all, also non-cardiac pathophysiologic disease could be classified in another way: pancreatitis can also have a lethal course). Please add more information on the classification of the conditions

- Thank you for this important comment. The classification was based on extensive discussions among experienced colleagues where we tried to reach a consensus. We totally agree with reviewer that for example, aortic stenosis and acute pancreatitis can present in different ways, but here we assume that abnormal vital signs may be helpful. Thus, abnormal vital signs may change the priority. Most likely, there will not be an absolute distinction between high and intermediate risk. Furthermore, there may be other reasons (for example frailty) which may force the EMS personnel to transport a patient with a low risk condition to hospital. We have added a few sentences about this to the discussion. Please, see the revised manuscript.

Minor Comment:

1. In table 1: A total AV block is no arrhythmia. Please change this description

- Thanks for this observation. This is now corrected.

2. If possible, a figure that illustrates the main results would be of additional value.

- A figure describing the main results have been added. Please see, figure 2.

3. What is the difference between: Undefined MI and MINOCA/MINCA?

- Undefined refers to patients diagnosed with AMI but it is not stated in the patient medical record which type of MI. This is now clarified in Table 1.

4. The lay-out of the tables could be improved. It is difficult to interpret the current extensive tables.

- Tables in the supplemental material layout have been revised. Tables in the manuscript have been slightly revised but will hopefully be further adjusted by BMJ open before eventual publication.

Reviewer: 3

Dr. Camila Caiado, Durham University

Comments to the Author:

The paper is quite interesting and warrants publication due to the value of the data collected and its presentation. I'm looking forward to seeing how this project progresses.

Discussion on generalization: The data collected is limited to a specific area in Sweden. If a risk score is to be developed, there has to be a discussion on if and how the study will expand to ensure the results are transferable to other parts of Sweden and beyond.

- We have expanded our discussion on generalisation. Please, see the revised manuscript.
- If upcoming results from model refining studies indicate that clinical usage would be appropriate validation multi-centre studies are planned to be carried out. Please, see the revised manuscript.

Population: I'd like to see a discussion on the socio-economical factors for the target population. What is the ethnic diversity in the target population and the observed cohort? Can other characteristics such as height and weight be included in this paper? If not, would it be possible to include in future publications?

- Unfortunately, we do not have access to data regarding height and weight. It would have been interesting to examine if these factors are of predictive importance. Maybe it is possible to consider this aspect in future studies.
- Unfortunately, we do not have specific data on socio-economic factors or ethnic diversity. We have expanded our discussion on this. Please, see the revised manuscript.

Response time: It would be interesting to know what the response time for these events were. What is the expected response time for EMS in this area? Were there any outliers?

- Data on response time with quartiles can be seen in supplemental material 3.

Pre-existing conditions: When looking at the risk factors considered, there is no clear distinction on which of those were diagnosed previously and/or are long-term conditions for the patient. would it be possible to reorganize the table to highlight known pre-existing conditions vs conditions diagnosed at the time? For a future publication, it would also useful to know when diagnosis for pre-existing conditions took place (e.g. less than 1 year, 1-5 years, etc.)

- All reported data on previous medical history concerns conditions diagnosed before the EMS mission. Unfortunately, we do not have data on exactly when the patients were diagnosed with the specific diagnoses in their previous medical history. This should be interesting to take into account in future studies as it may affect the prediction value of the previous medical history. A history of AMI one year ago compared to fifteen years ago probably have different predictive values. However, maybe it is difficult to implement the use of such detailed information in the prehospital emergency setting.

Missing data: I can't find any discussion on missing data. For a dataset of this size, there must have been a small proportion of missingness. Could the authors please add a statement to the paper? If no data was missing, great, please say so. If data was missing, please explain how it was addressed and whether any imputation techniques were used.

- Missing data for each included variable is presented in parentheses next to each variable in supplemental file 4 and 5.
- Missing data is also discussed in the strengths and limitation section. Please, see the revised manuscript.

AUC/ROC: There is a quick mention of the area under the curve for two of the risk groups with fairly similar AUCs. Would it be possible to present ROCs and/or expand the discussion in this area?

- ROC curves are now presented. Please, see figure 2. We also comment on this in the discussion section. Please, see the revised manuscript. However, the absence of a statistically significant discrepancy prevents conclusions beyond that both curves are quite similar.

VERSION 2 – REVIEW

REVIEWER	Collinson, Paul St Georges Hospital, Chemical Pathology
REVIEW RETURNED	18-Mar-2021

GENERAL COMMENTS	No further comments
---------------------

REVIEWER	Boogers, Mark Leiden Universitair Medisch Centrum, Cardiology
REVIEW RETURNED	18-Mar-2021

GENERAL COMMENTS	BMJ Open – Prehospital Stratification in Acute Chest Pain Patient into High and Low Risk By EMS – A prospective Cohort Study The current study describes a contemporary and interesting topic considering prehospital triage of chest pain patients. An appropriate selection of patients could potentially reduce unnecessary patient transfers or interclinical rides. Reduction of the total number of presented patients is of major interest due to overcrowding of emergency departments. The authors have used a well-structured prehospital data system, which still enables evaluation of risk models despite well-known difficulties in data acquisition in prehospital setting. Well-done. Comments: 1. The authors have identified many patient parameters for distinguishing high- and low-risk conditions, such as age and ST segment abnormalities. Multiple other factors were also associated with high risk conditions, including atrial ectopic beats or pain in the right arm, whereas classical factors such as cardiovascular risk factors or medical history were not related to these high risk conditions. This finding is probably related to the broad study endpoints (high risk conditions; ranging from PVE to ACS), rather than a small well-defined endpoint, such as ACS or
--

	NSTEMI/STEMI. For this reason, its important to mention that the current risk variables are able to discriminate patients for low- or high risk conditions (leave patient at home or transfer patient to PCI center), but selection for cardiac or non-cardiac presentations is more problematic. Such patients can not be send easily to a dedicated cardiac department directly, as important non-cardiac pathology could also be present as well. So, the study risk model is rather a general EMS risk model, rather than a cardiac prediction model. This is of importance, as some hospitals can see patients directly on their cardiac departments. 2. For study purposes, a wide-range of parameters have been selected to build a multivariate model. In clinical practice, one should aim to condense the prediction model as much as possible, in order to keep it user-friendly at the scene. Is there a way to shorten the list of patient parameters even more? please discuss. 3. In the intermediate risk conditions, aortic valve stenosis is reported. The presentations of aortic valve stenosis can differ from asymptomatic minor valve stenosis to even critical stenosis. Please discuss in more detail what kind of stenosis is meant. And what is the reason for only selection aortic valve stenosis, and no other major valve disease (severe mitral valve regurgitation)? 4. Could the authors please explain why conventional vital parameters (e.g. tachypnea, low blood pressure, tachycardia) were not associated with high-risk conditions? 5. Although EMS decisions for patient presentation to a hospital is largely depended to clinical status of the patients, one should probably take frailty (medical history, cognitive status) of the patient into account. What is the best care for this patient at this time? How do the authors think about this additional information? Please discuss. 6. The data of the study is based on patients form 2018. Why did the authors use these data instead of 2019/2020? One would assume that these data is also available Minor comment 1. What is difference between NSTEMI and 'undefined AMI'?
--	--

VERSION 2 – AUTHOR RESPONSE

Reviewer: 1

Dr. Paul Collinson, St Georges Hospital

Comments to the Author:

No further coments

Reviewer: 1

Competing interests of Reviewer: None declared

Reviewer: 2

Dr. Mark Boogers, Leiden Universitair Medisch Centrum

Comments to the Author:

BMJ Open – Prehospital Stratification in Acute Chest Pain Patient into High and Low Risk By EMS – A prospective Cohort Study

The current study describes a contemporary and interesting topic considering prehospital triage of chest pain patients. An appropriate selection of patients could potentially reduce unnecessary patient transfers or interclinical rides. Reduction of the total number of presented patients is of major interest due to overcrowding of emergency departments. The authors have used a well-structured prehospital data system, which still enables evaluation of risk models despite well-known difficulties in data acquisition in prehospital setting. Well-done.

Comments:

1. The authors have identified many patient parameters for distinguishing high- and low-risk conditions, such as age and ST segment abnormalities. Multiple other factors were also associated with high risk conditions, including atrial ectopic beats or pain in the right arm, whereas classical factors such as cardiovascular risk factors or medical history were not related to these high risk conditions. This finding is probably related to the broad study endpoints (high risk conditions; ranging from PVE to ACS), rather than a small well-defined endpoint, such as ACS or NSTEMI/STEMI. For this reason, its important to mention that the current risk variables are able to discriminate patients for low- or high risk conditions (leave patient at home or transfer patient to PCI center), but selection for cardiac or non-cardiac presentations is more problematic.

Such patients can not be send easily to a dedicated cardiac department directly, as important non-cardiac pathology could also be present as well. So, the study risk model is rather a general EMS risk model, rather than a cardiac prediction model. This is of importance, as some hospitals can see patients directly on their cardiac departments.

Agree! Our ambition is to develop a prediction model for prehospital risk assessment to use in order to improve patient triage and identify patients without need of ambulance transport to hospital not to identify cardiac patients specifically. Referring patients directly to a cardiac department should be done with great caution given that our high-risk endpoint includes several non-cardiac conditions. A section is added on this subject in the discussion section of the revised manuscript.

2. For study purposes, a wide-range of parameters have been selected to build a multivariate model. In clinical practice, one should aim to condense the prediction model as much as possible, in order to keep it user-friendly at the scene. Is there a way to shorten the list of patient parameters even more? please discuss.

Agree! In order to succeed in clinical implementation a prediction model should preferably be condensed, using a limited number of factors. The present manuscript aims to inform future risk assessment tools with capabilities of prehospital risk assessment. Derived models must by refined further before clinical testing. Missing data make such model refinement problematic. Especially since Troponin T is likely to be a cornerstone in such a refined model and the rates of missing for Troponin T is unfortunately quite high. However, this report is only the first to come, on the development of a decision support tool for the assessment of EMS patients with chest pain. With further research we hope to be able to present a more condensed prediction model. We have previously discussed this in the manuscript where the message of the reviewer was intended to be stressed. This paragraph is highlighted in the marked manuscript copy.

3. In the intermediate risk conditions, aortic valve stenosis is reported. The presentations of aortic valve stenosis can differ from asymptomatic minor valve stenosis to even critical stenosis. Please discuss in more detail what kind of stenosis is meant. And what is the reason for only selection aortic valve stenosis, and no other major valve disease (severe mitral valve regurgitation)?

Consensus on definitions of how to risk classify EMS patients is lacking. ¹ When conducting this study extensive discussions among experienced colleagues were carried out in order to reach a consensus on risk classification. This was validated by an external cardiologist. The Kappa coefficient showed substantial agreement between the risk classification between the authors and this external cardiologist. However, both the extensive discussion and the external validation show how problematic it is to define such risk classification groups. This was also pointed out in previous work of the authors. ¹ Regarding aortic valve stenosis we don't have data on what kind of stenosis the patient suffered from. We only used the ICD-10 code used to diagnose the patients on hospital discharge. We deemed it reasonable to classify aortic valve stenosis as intermediate risk since, if new, it should be investigated in hospital but the time factor is rarely crucial. We agree, that these patients should be transported to hospital. If severely affected by for example mitral valve insufficiency this will likely be manifested in abnormal vital signs prompting rapid transport to hospital. This is also described in the manuscript.

The reason for not including other valve diseases was that no patients with other major valve disease than aortic valve stenosis was identified in the study. A few patients with endocarditis were identified but data on whether these had valvular heart disease was not obtained. If there is a suspicion of valvular heart disease as the underlying ethology causing the chest pain it is reasonable to take the patient to hospital.

4. Could the authors please explain why conventional vital parameters (e.g. tachypnea, low blood pressure, tachycardia) were not associated with high-risk conditions?

This finding is somewhat surprising. However, this is not a unique finding for our study.² It may be explained by the rareness of deviating vital signs which affect the statistical prerequisites for identifying possible associations. The vast majority of patients in our cohort present with normal vital signs. Maybe the use of a larger cohort may identify such associations. However, the use of such rare factors for risk prediction is not unproblematic. We have expanded the discussion on this topic in the revised manuscript.

This finding stress the need to use other factors than vital signs for risk assessment of patients with chest pain in the prehospital setting.

However, as described in both the discussion and the "strengths and limitations" section we think that deviating vital signs in themselves are reasons enough for prompt hospital care. These paragraphs are high-lighted in the marked copy.

5. Although EMS decisions for patient presentation to a hospital is largely depended to clinical status of the patients, one should probably take frailty (medical history, cognitive status) of the patient into account. What is the best care for this patient at this time? How do the authors think about this additional information? Please discuss.

Agree! The EMS personnel cannot only consider strict medical factors when assessing patient care needs. Other, “softer” aspects must always be accounted for. We have expanded the discussion on this topic in the revised manuscript. This is also discussed in the “strengths and limitation” section. These paragraphs are high-lighted in the marked copy.

6. The data of the study is based on patients form 2018. Why did the authors use these data instead of 2019/2020? One would assume that these data is also available

This is explained by the fact that the process of obtaining the current data was very time consuming. We did not only use routinely collected data. For example all data regarding symptoms was collected using a specific questionnaire. Furthermore, the process of obtaining the diagnoses on discharge was not possible automatically by computerised data extraction but was manually collected in patients’ medical records. The work of cleaning and analysing collected data was also quite time consuming.

Minor comment

7. What is difference between NSTEMI and ‘undefined AMI’?

Undefined AMI refers to when the type of AMI was not stated in the patient’s medical record. This is explained in the footnote of Table 1.

Reviewer: 2

Competing interests of Reviewer: None declared

References

1. Wibring K, Magnusson C, Axelsson C, et al. Towards definitions of time-sensitive conditions in prehospital care. *Scand J Trauma Resusc Emerg Med* 2020;28(1):7. doi: 10.1186/s13049-020-0706-3 [published Online First: 2020/01/29]
2. Frisch A, Heidle KJ, Frisch SO, et al. Factors associated with advanced cardiac care in prehospital chest pain patients. *Am J Emerg Med* 2017 doi: 10.1016/j.ajem.2017.12.003 [published Online First: 2017/12/05]